# Reinforcement learning with Human Feedback: Learning Dynamic Choices via Pessimism

## Abstract

In this paper we study offline Reinforcement Learning with Human Feedback (RLHF) where we aim to learn the human's underlying reward and the MDP's optimal policy from a set of trajectories induced by human choices. We focus on the Dynamic Discrete Choice (DDC) model for modeling and understanding human choices, which is widely used to model a human decision-making process with forward-looking and bounded rationality. We propose a Dynamic-Choice-Pessimistic-Policy-Optimization (DCPPO) method and prove that the suboptimality of DCPPO *almost* matches the classical pessimistic offline RL algorithm in terms of suboptimality's dependency on distribution shift and dimension. To the best of our knowledge, this paper presents the first theoretical guarantees for off-policy offline RLHF with dynamic discrete choice model.

## 1. Introduction

*Reinforcement Learning with Human Feedback* (RLHF) is an area in machine learning research that incorporates human guidance or feedback to learn an optimal policy. In recent years, RLHF has achieved significant success in large language models, clinical trials, auto-driving, robotics, etc. (Ouyang et al., 2022; Gao et al., 2022; Glaese et al., 2022; Hussein et al., 2017; Jain et al., 2013; Kupcsik et al., 2018; Menick et al., 2022; Nakano et al., 2021; Novoseller et al., 2020). Unlike conventional offline reinforcement learning, where the learner aims to determine the optimal policy using observable reward data, in RLHF, the learner does not have direct access to the reward signal but instead can only observe a historical record of visited states and human-preferred actions. In such cases, the acquisition of reward

[1]Anonymous Institution, Anonymous City, Anonymous Region, Anonymous Country. Correspondence to: Anonymous Author <anon.email@domain.com>.

Preliminary work. Under review by the International Conference on Machine Learning (ICML). Do not distribute.

knowledge becomes pivotal.

*Dynamic Discrete Choice* (DDC) model is a framework for studying learning for human choices from data, which has been extensively studied in econometrics literature. (Rust, 1987; Hotz & Miller, 1993; Hotz et al., 1994; Aguirre-gabiria & Mira, 2002; Kalouptsidi et al., 2021; Bajari et al., 2015; Chernozhukov et al., 2022). In a DDC model, the agent make decisions under unobservable perturbation, i.e. $\pi_h(a_h \mid s_h) = \mathrm{argmax}_a\{Q_h(s_h, a) + \epsilon_h(a)\}$, where $\epsilon_h$ is an unobservable random noise and $Q_h$ is the agent's action value function.

In this work, we focus on RLHF within the context of a dynamic discrete choice model. Our challenges are three-folded: (i) The agent must first learn the human behavior policies from the feedback data. (ii) As the agent's objective is to maximize cumulative reward, the reward itself is not directly observable. We need to estimate the reward from the behavior policies. (iii) We face the challenge of insufficient dataset coverage and large state space.

With these coupled challenges, we ask the following question:

*Without access to the reward function, can one learn the optimal pessimistic policy from merely human choices under the dynamic choice model?*

**Our Results.** In this work, we propose the Dynamic-Choice-Pessimistic-Policy-Optimization (DCPPO) algorithm. By addressing challenges (i)-(iii), our contributions are three folds: (i) For learning behavior policies in large state spaces, we employ maximum likelihood estimation to estimate state/action value functions with function approximation. We establish estimation error bounds for general model class with low covering number. (ii) Leveraging the learned value functions, we minimize the Bellman mean squared error (BMSE) through linear regression. This allows us to recover the unobservable reward from the learned policy. Additionally, we demonstrate that the error of our estimated reward can be efficiently controlled by an uncertainty quantifier. (iii) To tackle the challenge of insufficient coverage, we follow *the principle of pessimism*, by incorporating a penalty into the value function during value iteration.

We establish the suboptimality of our algorithm with high probability with only single-policy coverage.

Our result matches existing pessimistic offline RL algorithms in terms of suboptimality's dependence on distribution shift and dimension, even in the absence of an observable reward. To the best of our knowledge, our results offer the first theoretical guarantee for pessimistic RL under the human dynamic choice model.

## 1.1. Related Work

**Reinforcement Learning with Human Feedback.** In recent years RLHF and inverse reinforcement learning (IRL) has been widely applied to robotics, recommendation system, and large language model (Ouyang et al., 2022; Lindner et al., 2022; Menick et al., 2022; Jaques et al., 2020; Lee et al., 2021; Nakano et al., 2021). However, there are various ways to incorporate human preferences or expertise into the decision-making process of an agent. (Shah et al., 2015; Ouyang et al., 2022; Saha & Krishnamurthy, 2022) learn reward from pairwise comparison and ranking. (Pacchiano et al., 2021) study pairwise comparison with function approximation in pairwise comparison. (Zhu et al., 2023) study various cases of preference-based-comparison in contextual bandit problem with linear function approximation, however convergence of their algorithm relies on the implicit assumption of sufficient coverage. The majority of prior researches in RLHF only consider bandit cases and have not studied MDP case with transition dynamics. (Wang et al., 2018) study how to learn a uniformly better policy of an MDP from an offline dataset by learning the advantage function. However, they cannot guarantee the learned policy converges to the optimal policy.

**Dynamic Discrete Choice Model.** Dynamic Discrete Choice (DDC) model is a widely studied choice model in econometrics and is closely related to reward learning in IRL and RLHF. In the DDC model, the human agent is assumed to make decisions under the presence of Gumbel noise (Type I Extreme Error)(Aguirregabiria & Mira, 2002; Chernozhukov et al., 2022; Bajari et al., 2015; Kalouptsidi et al., 2021; Adusumilli & Eckardt, 2019), i.e. under bounded rationality, and the task is to infer the underlying utility. Most work in econometrics cares for asymptotic $\sqrt{n}$-convergence of estimated utility, and does not study finite sample estimation error. Moreover, their methods suffer from significant computation burdens from large or high dimensional state space (Zeng et al., 2022). In recent years, there has been work combining the dynamic discrete choice model and IRL. (Zeng et al., 2022) prove the equivalence between DDC estimation problem and maximum likelihood IRL problem and propose an online gradient method for reward estimation under ergodic

dynamics assumption. (Zeng et al., 2023) reformulate the reward estimation in the DDC model into a bilevel optimization and propose a model-based approach by assuming an environment simulator.

**Offline Reinforcement Learning and Pessimism.** The idea of introducing pessimism for offline RL to deal with distribution shift has been studied in recent years (Jin et al., 2021; Uehara et al., 2021). (Jin et al., 2021) show that pessimism is sufficient to eliminate spurious correlation and intrinsic uncertainty when doing value iteration. (Uehara et al., 2021) show that with single-policy coverage, i.e. coverage over the optimal policy, pessimism is sufficient to guarantee a $\mathcal{O}(n^{-1/2})$ suboptimality. In this paper, we connect RLHF with offline RL and show our algorithm achieves pessimism by designing an uncertainty quantifier that can tackle error from estimating reward functions, which is crucial in pessimistic value iteration.

## 1.2. Notations and Preliminaries

For a positive-semidefinite matrix $A \in \mathbb{R}^{d \times d}$ and vector $x \in \mathbb{R}^d$, we use $\|x\|_A$ to denote $\sqrt{x^\top A x}$. For an arbitrary space $\mathcal{X}$, we use $\Delta(\mathcal{X})$ to denote the set of all probability distribution on $\mathcal{X}$. For two vectors $x, y \in \mathbb{R}^d$, we denote $x \cdot y = \sum_i^d x_i y_i$ as the inner product of $x, y$. We denote the set of all probability measures on $\mathcal{X}$ as $\Delta(\mathcal{X})$. We use $[n]$ to represent the set of integers from 0 to $n-1$. For two matrices $A$ and $B$, we write $A \succeq B$ if $A - B \succeq 0$. We define a finite horizon MDP model $M = (\mathcal{S}, \mathcal{A}, H, \{P_h\}_{h \in [H]}, \{r_h\}_{h \in [H]})$, $H$ is the horizon length, in each step $h \in [H]$, the agent starts from state $s_h$ in the state space $\mathcal{S}$, chooses an action $a_h \in \mathcal{A}$ with probability $\pi_h(a_h \mid s_h)$, receives a reward of $r_h(s_h, a_h)$ and transits to the next state $s'$ with probability $P_h(s' \mid s_h, a_h)$. Here $\mathcal{A}$ is a finite action set with $|\mathcal{A}|$ actions and $P_h(\cdot | s_h, a_h) \in \Delta(s_h, a_h)$ is the transition kernel condition on state action pair $(s, a)$. For convenience we assume that $r_h(s, a) \in [0, 1]$ for all $(s, a, h) \in \mathcal{S} \times \mathcal{A} \times [H]$. Without loss of generality, we assume that the initial state of each episode $s_0$ is fixed. Note that this will not add difficulty to our analysis. For any policy $\pi = \{\pi_h\}_{h \in [H]}$ the state value function is $V_h^\pi(s) = \mathbb{E}_\pi\left[\sum_{t=h}^H r_t(s_t, a_t) \mid s_h = s\right]$, and the action value function is $Q_h^\pi(s, a) = \mathbb{E}_\pi\left[\sum_{t=h}^H r_t(s_t, a_t) \mid s_h = s, a\right]$, here the expectation $\mathbb{E}_\pi$ is taken with respect to the randomness of the trajectory induced by $\pi$, i.e. is obtained by taking action $a_t \sim \pi_t(\cdot \mid s_t)$ and observing $s_{t+1} \sim P_h(\cdot \mid s_t, a_t)$. For any function $f : \mathcal{S} \to \mathbb{R}$, we define the transition operator $\mathbb{P}_h f(s, a) = \mathbb{E}[f(s_{h+1}) \mid s_h = s, a_h = a]$. We also define the Bellman equation for any policy $\pi$, $V_h^\pi(s) = \langle \pi_h(a \mid s), Q_h^{\pi_b}(s, a) \rangle, Q_h^\pi(s, a) = r_h(s, a) + \mathbb{P}_h V_{h+1}^\pi(s, a)$. For an MDP we denote its optimal policy as $\pi^*$, and define the performance metric for any

policy $\pi$ as $\mathrm{SubOpt}(\pi) = V_1^{\pi^*} - V_1^\pi$.

## 2. Problem Formulation

In this paper, we aim to learn from a dataset of human choices under dynamic discrete choice model. Suppose we are provided with dataset $\mathcal{D} = \{\mathcal{D}_h = \{s_h^i, a_h^i\}_{i \in [n]}\}_{h \in [H]}$, containing $n$ trajectories collected by observing a single human behavior in a dynamic discrete choice model. Our goal is to learn the optimal policy $\pi^*$ of the underlying MDP. We assume that the agent is bounded-rational and makes decisions according to the dynamic discrete choice model (Rust, 1987; Hotz & Miller, 1993; Chernozhukov et al., 2022; Zeng et al., 2023). In dynamic discrete choice model, the agent's policy has the following characterization (Rust, 1987; Aguirregabiria & Mira, 2002; Chernozhukov et al., 2022),

$$\pi_{b,h}(a \mid s) = \frac{\exp(Q_h^{\pi_b,\gamma}(s,a))}{\sum_{a' \in \mathcal{A}} \exp(Q_h^{\pi_b,\gamma}(s,a'))}, \qquad (1)$$

here $Q_h^{\pi_b,\gamma}(\cdot, \cdot)$ works as the solution of the discounted Bellman equation,

$$V_h^{\pi_b,\gamma}(s) = \langle \pi_{b,h}(a \mid s), Q_h^{\pi_b,\gamma}(s,a) \rangle, \qquad (2)$$

$$Q_h^{\pi_b,\gamma}(s,a) = r_h(s,a) + \gamma \cdot \mathbb{P}_h V_{h+1}^{\pi_b,\gamma}(s,a) \qquad (3)$$

for all $(s,a) \in \mathcal{S} \times \mathcal{A}$. Note that (2) differs from the original Bellman equation due to the presence of $\gamma$, which is a discount factor in $[0,1]$, and measures the myopia of the agent. The case of $\gamma = 0$ corresponds to a *myopic* human agent. Such choice model comes from the perturbation of noises,

$$\pi_{b,h}(\cdot \mid s_h) =$$
$$\mathrm{argmax}_{a \in \mathcal{A}} \left\{ r_h(s_h, a) + \epsilon_h(a) + \gamma \cdot \mathbb{P}_h V_{h+1}^{\pi_b,\gamma}(s_h, a) \right\},$$

where $\{\epsilon_h(a)\}_{a \in \mathcal{A}}$ are i.i.d Gumbel noises that is observed by the agent but not the learner, $\{V_h^{\gamma,\pi_b}\}_{h \in [H]}$ is the value function of the agent, and is widely used to model human decision. We also remark that the state value function defined in (2) corresponds to the *ex-ante* value function in econometric studies (Aguirregabiria & Mira, 2010; Arcidiacono & Ellickson, 2011; Bajari et al., 2015). When considering Gumbel noise as part of the reward, the value function may have a different form. However, such a difference does not add complexity to our analysis.

## 3. Reward Learning from Human Dynamic Choices

In this section, we present a general framework of an offline algorithm for learning the reward of the underlying MDP.

Our algorithm consists of two steps: (i) The first step is to estimate the agent behavior policy from the pre-collected dataset $\mathcal{D}$ by maximum likelihood estimation (MLE). We recover the action value functions $\{Q_h^{\pi_b,\gamma}\}_{h \in [H]}$ from (1) and the state value functions $\{V_h^{\pi_b,\gamma}\}_{h \in [H]}$ from (2) using function approximation. In Section 3.1, we analyze the error of our estimation and prove that for any model class with a small covering number, the error from MLE estimation is of scale $\tilde{\mathcal{O}}(1/n)$ in dataset distribution. We also remark that our result does not need the dataset to be well-explored, which is implicitly assumed in previous works (Zhu et al., 2023; Chen et al., 2020). (ii) We recover the underlying reward from the model class by minimizing a penalized Bellman MSE with plugged-in value functions learned in step (i). In Section 3.2, we study linear model MDP as a concrete example. Theorem 3.5 shows that the error of estimated reward can be bounded by an elliptical potential term for all $(s,a) \in \mathcal{S} \times \mathcal{A}$ in both settings. First, we make the following assumption for function approximation.

**Assumption 3.1** (**Function Approximation Model Class**). We assume the existence of a model class $\mathcal{M} = \{\mathcal{M}_h\}_{h \in [H]}$ containing functions $f : \mathcal{S} \times \mathcal{A} \to [0, H]$ for every $h \in [H]$, and is rich enough to capture $r_h$ and $Q_h$, i.e. $r_h \in \mathcal{M}_h, Q_h \in \mathcal{M}_h$.

In practice, $\mathcal{M}_h$ can be a (pre-trained) neural network or a random forest. We now present our algorithm for reward learning in RLHF.

---

**Algorithm 1** DCPPO: Reward Learning for General Model Class

---

**Require:** Dataset $\left\{ \mathcal{D}_h = \{s_h^i, a_h^i\}_{i \in [n]} \right\}_{h \in [H]}$, constant $\lambda > 0$, penalty function $\rho(\cdot)$, parameter $\beta$.
1: **for** step $h = H, \ldots, 1$ **do**
2:  Set $\widehat{Q}_h$ by maximizing (4).
3:  Set $\widehat{\pi}_h(a_h|s_h)$ by (5).
4:  Set $\widehat{V}_h(s_h) = \langle \widehat{Q}_h(s_h, \cdot), \widehat{\pi}_h(\cdot \mid s_h) \rangle_{\mathcal{A}}$.
5:  Set $\widehat{r}_h(s_h, a_h)$ by (6).
6: **end for**
7: **Output:** $\{\widehat{r}_h\}_{h \in [H]}$.

---

### 3.1. First Step: Recovering Human Policy and Human State-Action Values

For every step $h$, we use maximum liklihood estimaton (MLE) to estimate the behaviour policy $\pi_{b,h}$, corresponds to $Q_h^{\pi_b,\gamma}(s,a)$ in a general model class $\mathcal{M}_h$. For each step $h \in [H]$, we have the log-likelihood function

$$L_h(Q) = \frac{1}{n} \sum_{i=1}^{n} \log \left( \frac{\exp(Q(s_h^i, a_h^i))}{\sum_{a' \in \mathcal{A}} \exp(Q(s, a'))} \right) \qquad (4)$$

for $Q \in \mathcal{M}_h$, and we estimate $Q_h$ by maximizing (4). Then we recover the policy $\widehat{\pi}_h$ by

$$\widehat{\pi}_h(a_h \mid s_h) = \exp(\widehat{Q}_h(s_h, a_h)) / \sum_{a' \in \mathcal{A}} \exp(\widehat{Q}_h(s_h, a')). \tag{5}$$

Note that by Equation (1), adding a constant on $Q_h^{\pi_b, \gamma}$ will produce the same policy under dynamic discrete model, and thus the real behavior value function is unidentifiable in general. For identification, we have the following assumption.

**Assumption 3.2** (**Model Identification**). We assume that there exists one $a_0 \in \mathcal{A}$, such that $Q(s, a_0) = 0$ for every $s \in \mathcal{S}$.

Note that this assumption does not affect our further analysis. Other identifications includes parameter constraint (Zhu et al., 2023) or utility constraints (Bajari et al., 2015). We can ensure the estimation of the underlying policy and corresponding value function is accurate in the states the agent has encountered. Formally, we have the following theorem,

**Theorem 3.3** (**Policy and Value Functions Recovery from Choice Model**). *With Algorithm 1 , we have*

$$\mathbb{E}_{\mathcal{D}_h} \left[ \| \widehat{\pi}_h(\cdot \mid s_h) - \pi_{b,h}(\cdot \mid s_h) \|_1^2 \right]$$
$$\leq \mathcal{O} \left( \frac{\log \left( N(\mathcal{M}_h, \| \cdot \|_\infty, 1/n)/\delta \right)}{n} \right)$$

*and*

$$\mathbb{E}_{\mathcal{D}_h} \left[ \| \widehat{Q}_h(s_h, \cdot) - Q_h^{\pi_b, \gamma}(s_h, \cdot) \|_1^2 \right]$$
$$\leq \mathcal{O} \left( \frac{H^2 \cdot e^H \cdot \log \left( N(\mathcal{M}_h, \| \cdot \|_\infty, 1/n)/\delta \right)}{n} \right)$$

*with probability at least $1 - \delta$. Here $\mathbb{E}_{\mathcal{D}_h}[\cdot]$ means the expectation is taken on collected dataset $\mathcal{D}_h$, i.e. the mean value taken with respect to $\{s_h^i\}_{i \in [n]}$.*

Theorem 3.3 shows that we can efficiently learn $\pi_{b,h}$ from the dataset under identification assumption. As a result, we can provably recover the value functions by definition in Equation 1.

### 3.2. Reward Learning from Dynamic Choices

Bellman equation motivates the following estimate of the reward function:

$$\widehat{r}_h(s_h, a_h) = \tag{6}$$
$$\text{argmin}_{r \in \mathcal{M}_h} \Big\{ \sum_{i=1}^n \big( r_h(s_h^i, a_h^i) + \gamma \cdot \widehat{V}_{h+1}(s_{h+1}^i)$$
$$- \widehat{Q}_h(s_h^i, a_h^i) \big)^2 + \lambda \rho(r) \Big\},$$

i.e. we can recover the reward with previously learned $\widehat{V}_h, \widehat{Q}_h$ by minimizing Bellman MSE. As a concrete example, we study the instantiation of Algorithm 1 for the linear

model class. We define the function class $\mathcal{M}_h = \{f(\cdot) = \phi(\cdot)^\top \theta : \mathcal{S} \times \mathcal{A} \to \mathbb{R}, \theta \in \Theta\}$ for $h \in [H]$, where $\phi \in \mathbb{R}^d$ is the feature defined on $\mathcal{S} \times \mathcal{A}$, $\Theta$ is a subset of $\mathbb{R}^d$ which parameterizes the model class, and $d > 0$ is the dimension of the feature. Corresponding to Assumption 3.2, We also assume that $\phi(s, a_0) = 0$ for every $s \in \mathcal{S}$. Note that this model class contains the reward $r_h$ and state action value function $Q_h$ in tabular MDP where $\phi(s, a)$ is the one-hot vector of $(s, a)$. The linear model class also contains linear MDP, which assumes both the transition $P(s_{h+1} \mid s_h, a_h)$ and the reward $r_h(s_h, a_h)$ are linear functions of feature $\phi(s_h, a_h)$ (Jin et al., 2020; Duan et al., 2020; Jin et al., 2021). In linear model case, our first step MLE in (4) turns into a logistic regression,

$$\widehat{\theta}_h = \text{argmax}_{\theta \in \Theta} \frac{1}{n} \sum_{i=1}^n \phi(s_h^i, a_h^i) \cdot \theta - \log \left( \sum_{a' \in \mathcal{A}} \exp(\phi(s_h^i, a') \cdot \theta) \right), \tag{7}$$

which can be efficiently solved by existing state-of-art optimization methods. We now have $\{\widehat{Q}_h\}_{h \in [H]}, \{\widehat{\pi}_h\}_{h \in [H]}$ and $\{\widehat{V}_h\}_{h \in [H]}$ in Algorithm 1 to be our estimations for $Q_h^{\pi_b, \gamma}, \pi_{b,h}$ and $V_h^{\pi_b, \gamma}$ correspondingly.

Note that in linear case, Line 5 has a closed form solution,

$$\widehat{w}_h = (\Lambda_h + \lambda I)^{-1} \left( \sum_{i=1}^n \phi(s_h^i, a_h^i) \big( \widehat{Q}_h(s_h^i, a_h^i) - \gamma \cdot \widehat{V}_{h+1}(s_{h+1}^i) \big) \right), \tag{8}$$

$$\text{where } \Lambda_h = \sum_{i=1}^n \phi(s_h^i, a_h^i) \phi(s_h^i, a_h^i)^\top,$$

and we set $\widehat{r}(s_h, a_h) = \phi(s_h, a_h) \cdot \widehat{w}_h$. We also make the following assumption on the model class $\Theta$ and the feature function.

**Assumption 3.4** (**Regular Conditions**). We assume that: (i) For all $\theta \in \Theta$, we have $\|\theta\|_2 \leq \sqrt{d}$; (ii) For all $(s_h, a_h) \in \mathcal{S} \times \mathcal{A}$, $\|\phi(s_h, a_h)\|_2 \leq \sqrt{d}$. (iii)For all $n > 0$, $\log N(\Theta, \| \cdot \|_\infty, 1/n) \leq c \cdot d \log n$ for some absolute constant $c$.

We are now prepared to highlight our main result:

**Theorem 3.5** (**Reward Estimation for Linear Model MDP**). *With probability at least $1 - \delta$, we have the following estimation of our reward function for all $(s, a) \in \mathcal{S} \times \mathcal{A}$ and $\lambda > 0$,*

$$|r_h(s, a) - \widehat{r}_h(s, a)| \tag{9}$$
$$\leq \|\phi(s, a)\|_{(\Lambda_h + \lambda I)^{-1}}$$
$$\cdot \mathcal{O} \left( \sqrt{\lambda \cdot d} + (1 + \gamma) H e^H \cdot d \sqrt{\log \left( nH/\delta \right)} \right).$$

Note that the error can be bounded by the product of two terms, the elliptical potential term $\|\phi(s, a)\|_{(\Lambda + \lambda \cdot I)^{-1}}$ and the norm of a self normalizing term of scale $O(H e^H \cdot$

$d\sqrt{\log(n/\delta)})$. Here the exponential dependency $\mathcal{O}(e^H)$ comes from estimating $Q_h^{\pi_b,\gamma}$ with logistic regression and also occurs in logistic bandit (Zhu et al., 2023; Fei et al., 2020). It remains an open question if this additional factor can be improved, and we leave it for future work.

*Remark* 3.6. We remark that except for the exponential term in $H$, Theorem 3.5 *almost* matches the result when doing linear regression on an observable reward dataset, in which case error of estimation is of scale $\tilde{\mathcal{O}}(\|\phi(s,a)\|_{(\Lambda+\lambda I)^{-1}} \cdot dH)$ (Ding et al., 2021; Jin et al., 2021). When the human behavior policy has sufficient coverage, i.e. the minimal eigenvalue of $\mathbb{E}_{\pi_b}[\phi\phi^\top]$, $\sigma_{\min}(\mathbb{E}_{\pi_b}[\phi\phi^\top]) \geq 0$, we have $\|\phi(s,a)\|_{(\Lambda_h+\lambda I)^{-1}} = \mathcal{O}(n^{-1/2})$ holds for all $(s,a) \in \mathcal{S} \times \mathcal{A}$ (Duan et al., 2020) and $\|r_h - \hat{r}_h\|_\infty = \mathcal{O}(n^{-1/2})$.

# 4. Policy Learning from Dynamic Choices via Pessimistic Value Iteration

In this section, we describe the pessimistic value iteration algorithm, which minus a penalty function $\Gamma_h : \mathcal{S} \times \mathcal{A} \to \mathbb{R}$ from the value function when choosing the best action. Pessimism is achieved when $\Gamma_h$ is a *uncertainty quantifier* for our learned value functions $\{\tilde{V}_h\}_{h\in[H]}$, i.e. $\left|(\hat{r}_h + \widetilde{\mathbb{P}}_h\tilde{V}_{h+1})(s,a) - (r_h + \mathbb{P}_h\tilde{V}_{h+1})(s,a)\right| \leq \Gamma_h(s,a)$ for all $(s,a) \in \mathcal{S} \times \mathcal{A}$ with high probability. Then we use $\{\Gamma_h\}_{h\in[H]}$ as the penalty function for pessimistic planning, which leads to a conservative estimation of the value function. We formally describe our planning method in Algorithm 2. However, when doing pessimistic value iteration with $\{\hat{r}_h\}_{h\in[H]}$ learned from human feedback, it is more difficult to design uncertainty quantifiers, since the estimation error from reward learning is inherited in pessimistic planning. In Section 4.1, we propose an efficient uncertainty quantifier and prove that with pessimistic value iteration, Algorithm 2 can achieve a $\mathcal{O}(n^{-1/2})$ suboptimality gap.

---

**Algorithm 2** DCPPO: Pessimistic Value iteration

---

**Require:** Surrogate reward $\{\hat{r}_h(\cdot,\cdot)\}_{h\in[H]}$, collected dataset $\{(s_h^i, a_h^i)\}_{i\in[n],h\in[H]}$, parameter $\beta$, penalty .
1: Set $\tilde{V}_{H+1}(\cdot) = 0$.
2: **for** step $h = H, \ldots, 1$ **do**
3:     Set $\widetilde{\mathbb{P}}_h\tilde{V}_{h+1}(s_h, a_h)$ by (10).
4:     Construct $\Gamma_h(s_h, a_h)$ based on $\mathcal{D}$.
5:     Set $\tilde{Q}_h(s_h, a_h) = \min\{\hat{r}_h(s_h, a_h) + \widetilde{P}_h\tilde{V}_{h+1}(s_h, a_h) - \Gamma_h(s_h, a_h), H - h + 1\}_+$.
6:     Set $\tilde{\pi}_h(\cdot \mid \cdot) = \operatorname{argmax}\langle\tilde{Q}_h(\cdot,\cdot), \pi_h(\cdot \mid \cdot)\rangle$.
7:     Set $\tilde{V}_h = \langle\tilde{Q}_h(\cdot,\cdot), \tilde{\pi}_h(\cdot \mid \cdot)\rangle$.
8: **end for**
9: **Output:** $\{\tilde{\pi}_h\}_{h\in[H]}$.

---

## 4.1. Suboptimality Gap of Pessimitic Optimal Policy

In Line (3) of Algorithm 2, we update $\mathbb{P}_h\hat{V}_{h+1}$ by solving the following minimization:

$$\widetilde{\mathbb{P}}_h\tilde{V}_{h+1}(s_h, a_h) = \operatorname{argmin}_{f\in\mathcal{M}} \sum_{i\in[n]} \left(f(s_h^i, a_h^i) - \tilde{V}_{h+1}(s_{h+1})\right)^2. \tag{10}$$

For linear model class defined in Section 3.2, we assume that we can capture the conditional expectation of value function in the next step with the known feature $\phi$. In formal words, we make the following assumption.

**Assumption 4.1** (**Linear MDP**). For the underlying MDP, we assume that for every $V_{h+1} : \mathcal{S} \to [0, H-h]$, there exists $u_h \in \mathbb{R}^d$ such that

$$\mathbb{P}_h V_{h+1}(s,a) = \phi(s,a) \cdot u_h$$

for all $(s,a) \in \mathcal{S} \times \mathcal{A}$. We also assume that $\|u_h\| \leq (H - h + 1) \cdot \sqrt{d}$ for all $h \in [H]$.

Note that this assumption is directly satisfied by linear MDP class (Jin et al., 2021; 2020; Yang & Wang, 2019). For linear model MDP defined in Section 3.2, it suffices to have the parameter set $\Theta$ being closed under subtraction, i.e. if $x, y \in \Theta$ then $x - y \in \Theta$. Meanwhile, we construct $\Gamma_h$ in Algorithm 2 based on dataset $\mathcal{D}$ as

$$\Gamma_h(s,a) = \beta \cdot \left(\phi(s,a)^\top (\Lambda_h + \lambda I)^{-1} \phi(s,a)\right)^{1/2} \tag{11}$$

for every $h \in [H]$. Here that $\Lambda_h$ is defined in (8). To establish suboptimality for Algorithm 2, we assume that the trajectory induced by $\pi^*$ is "covered" by $\mathcal{D}$ sufficiently well.

**Assumption 4.2** (**Single-Policy Coverage**). Suppose there exists an absolute constant $c^\dagger > 0$ such that

$$\Lambda_h \geq c^\dagger \cdot n \cdot \mathbb{E}_{\pi^*}\left[\phi(s_h, a_h)\phi(s_h, a_h)^\top\right]$$

holds with probability at least $1 - \delta/2$.

We remark that Assumption 4.2 only assumes the human behavior policy can cover the optimal policy and is therefore weaker than assuming well-explored dataset, or sufficient coverage(Duan et al., 2020; Jin et al., 2021). With this assumption, we prove the following theorem.

**Theorem 4.3** (**Suboptimality Gap for DCPPO**). *Suppose Assumption 3.2, 3.4, 4.1, 4.2 holds. With $\lambda = 1$ and $\beta = \mathcal{O}(He^H \cdot d\sqrt{\log(nH/\delta)})$, we have (i) $\Gamma_h$ defined in (11) being uncertainty quantifiers, and (ii)*

$$\operatorname{SubOpt}\left(\{\tilde{\pi}_h\}_{h\in[H]}\right) \leq c \cdot (1+\gamma)d^{3/2}H^2 e^H n^{-1/2}\sqrt{\xi}$$

*holds with probability at least $1 - \delta$, here $\xi = \log(dHn/\delta)$. In particular, if $\operatorname{rank}(\Sigma_h) \leq r$ at each step $h \in [H]$, then*

$$\operatorname{SubOpt}\left(\{\tilde{\pi}_h\}_{h\in[H]}\right) \leq c \cdot (1+\gamma)r^{1/2}dH^2 e^H n^{-1/2}\sqrt{\xi},$$

*here $\Sigma_h = \mathbb{E}_{\pi_b}[\phi(s_h, a_h)\phi(s_h, a_h)^\top]$.*

**Remark.** It is worth highlighting that Theorem 4.3 nearly matches the standard result for pessimistic offline RL with observable rewards in terms of the dependence on data size and distribution, up to a constant factor of $\mathcal{O}(He^H)$ (Jin et al., 2020; Uehara & Sun, 2021), where their suboptimality is of $\tilde{\mathcal{O}}(dH^2n^{-1/2})$. Therefore, Algorithm 1 and 2 *almost* matches the suboptimality gap of standard pessimism planning with an observable reward, except for a $\mathcal{O}(e^H)$ factor inherited from reward estimation.

# 5. DCPPO for Reproducing Kernel Hilbert Space

In this section, we assume the model class $\mathcal{M} = \{\mathcal{M}_h\}_{h \in [H]}$ are subsets of a Reproducing Kernel Hilbert Space (RKHS). For notations simplicity, we let $z = (s, a)$ denote the state-action pair and denote $\mathcal{Z} = \mathcal{S} \times \mathcal{A}$ for any $h \in [H]$. We view $\mathcal{Z}$ as a compact subset of $\mathbb{R}^d$ where the dimension $d$ is fixed. Let $\mathcal{H}$ be an RKHS of functions on $\mathcal{Z}$ with kernel function $K : \mathcal{Z} \times \mathcal{Z} \to \mathbb{R}$, inner product $\langle \cdot, \cdot \rangle : \mathcal{H} \times \mathcal{H} \to \mathbb{R}$ and RKHS norm $\| \cdot \|_{\mathcal{H}} : \mathcal{H} \to \mathbb{R}$. By definition of RKHS, there exists a feature mapping $\phi : \mathcal{Z} \to \mathcal{H}$ such that $f(z) = \langle f, \phi(z) \rangle_{\mathcal{H}}$ for all $f \in \mathcal{H}$ and all $z \in \mathcal{Z}$. Also, the kernel function admits the feature representation $K(x, y) = \langle \phi(x), \phi(y) \rangle_{\mathcal{H}}$ for any $x, y \in \mathcal{H}$. We assume that the kernel function is uniformly bounded as $\sup_{z \in \mathcal{Z}} K(z, z) < \infty$.

Let $\mathcal{L}^2(\mathcal{Z})$ be the space of square-integrable functions on $\mathcal{Z}$ and let $\langle \cdot, \cdot \rangle_{\mathcal{L}^2}$ be the inner product for $\mathcal{L}^2(\mathcal{Z})$. We define the Mercer operater $T_K : \mathcal{L}^2(\mathcal{Z}) \to \mathcal{L}^2(\mathcal{Z})$,

$$T_K f(z) = \int_{\mathcal{Z}} K(z, z') \cdot f(z') \, \mathrm{d}z', \quad \forall f \in \mathcal{L}^2(\mathcal{Z}). \quad (12)$$

By Mercer's Theorem (Steinwart & Christmann, 2008), there exists a countable and non-increasing sequence of non-negative eigenvalues $\{\sigma_i\}_{i \geq 1}$ for the operator $T_K$, and the associated eigenfunctions $\{\psi_i\}_{i > 1}$ form an orthogonal basis of $\mathcal{L}^2(\mathcal{Z})$. In what follows, we assume the eigenvalue of the integral operator defined in 12 has a certain decay condition.

**Assumption 5.1** (Eigenvalue Decay of $\mathcal{H}$). Let $\{\sigma_j\}_{j \geq 1}$ be the eigenvalues induced by the integral opretaor $T_K$ defined in Equation (12) and $\{\psi_j\}_{j \geq 1}$ be the associated eigenfunctions. We assume that $\{\sigma_j\}_{j \geq 1}$ satisfies one of the following conditions for some constant $\gamma > 0$.

(i) $\mu$-finite spectrum: $\sigma_j = 0$ for all $j > \mu$, where $\mu$ is a positive integer.

(ii) $\mu$-exponential decay: there exists some constants $C_1, C_2 > 0, \tau \in [0, 1/2)$ and $C_\psi > 0$ such that $\sigma_j \leq C_1 \cdot \exp(-C_2 \cdot j^\mu)$ and $\sup_{z \in \mathcal{Z}} \sigma_j^\tau \cdot |\psi_j(z)| \leq C_\psi$ for all $j \geq 1$.

(iii) $\mu$-polynomial decay: there exists some constants $C_1 > 0, \tau \in [0, 1/2)$ and $C_\psi > 0$ such that $\sigma_j \leq C_1 \cdot j^{-\mu}$ and $\sup_{z \in \mathcal{Z}} \sigma_j^\tau \cdot |\psi_j(z)| \leq C_\psi$ for all $j \geq 1$, where $\mu > 1$.

For a detailed discussion of eigenvalue decay in RKHS, we refer the readers to Section 4.1 of (Yang et al., 2020).

## 5.1. Gurantee for RKHS

In RKHS case, our first step MLE in (4) turns into a kernel logistic regression,

$$\bar{Q}_h = \mathrm{argmin}_{Q \in \mathcal{H}} \frac{1}{n} \sum_{i=1}^n Q(s_h^i, a_h^i) - \log\left(\sum_{a' \in \mathcal{A}} \exp(Q(s, a'))\right). \quad (13)$$

Line 5 in Algorithm 1 now turns into a kernel ridge regression for the Bellman MSE, with $\rho(f)$ being $\|f\|_{\mathcal{H}}^2$. Following Representer's Theorem (Steinwart & Christmann, 2008), we have the following closed form solution

$$\widehat{r}_h(z) = k_h(z)^\top (K_h + \lambda \cdot I)^{-1} y_h,$$

where we define the Gram matrix $K_h \in \mathbb{R}^{n \times n}$ and the function $k_h : \mathcal{Z} \to \mathbb{R}^n$ as

$$K_h = \left[K(z_h^i, z_h^{i'})\right]_{i, i' \in [n]} \in \mathbb{R}^{n \times n}, \quad k_h(z) = \left[K(z_h^i, z)\right]_{i \in [n]}^\top \in \mathbb{R}^n, \quad (14)$$

and the entry of the response vector $y_h \in \mathbb{R}^n$ corresponding to $i \in [n]$ is

$$[y_h]_i = \widehat{Q}_h(s_h^i, a_h^i) - \gamma \cdot \widehat{V}_{h+1}(s_{h+1}^i).$$

Meanwhile, we also construct the uncertainty quantifier $\Gamma_h$ in Algorithm 2,

$$\Gamma_h(z) = \beta \cdot \lambda^{-1/2} \cdot \left(K(z, z) - k_h(z)^\top (K_h + \lambda I)^{-1} k_h(z)\right)^{1/2}$$

for all $z \in \mathcal{Z}$. Parallel to Assumption 4.1, we impose the following structural assumption for the kernel setting.

**Assumption 5.2.** Let $R_r > 0$ be some fixed constant and we define function class $\mathcal{Q} = \{f \in \mathcal{H} : \|f\|_{\mathcal{H}} \leq HR_r\}$. We assume that $\mathbb{P}_h V_{h+1} \in \mathcal{Q}$ for any $V_{h+1} : \mathcal{S} \to [0, H]$. We also assume that $\|r\|_{\mathcal{H}} \leq R_r$ for some constant $R_r > 0$. We set the model class $\mathcal{M}_h = \mathcal{Q}$ for all $h \in [H]$.

The above assumption states that the Bellman operator maps any bounded function into a bounded RKHS-norm ball, and holds for the special case of linear MDP (Jin et al., 2021).

Besides the closeness assumption on the Bellman operator, we also define the maximal information gain (Srinivas et al., 2009) as a characterization of the complexity of $\mathcal{H}$ :

$$G(n, \lambda) = \sup\{1/2 \cdot \log \det(I + K_{\mathcal{C}}/\lambda) : \mathcal{C} \subset \mathcal{Z}, |\mathcal{C}| \leq n\} \quad (15)$$

Here $K_{\mathcal{C}}$ is the Gram matrix for the set $\mathcal{C}$, defined similarly as Equation (14). In particular, when $\mathcal{H}$ has $\mu$-finite spectrum, $G(n, \lambda) = \mathcal{O}(\mu \cdot \log n)$ recovers the dimensionality of the linear space up to a logarithmic factor.

We are now ready to present our result for reward estimation in the RKHS case.

**Theorem 5.3 (Reward Estimation for RKHS).** *For Algorithm 1 and 2, with probability at least $1 - \delta$, we have the following estimations of our reward function for all $z \in \mathcal{Z} \times \mathcal{A}$ and $\lambda \geq 1 + 1/n$,*

$$
\begin{aligned}
&|r_h(z) - \widehat{r}_h(z)| \\
&\quad \leq \|\phi(z)\|_{(\Lambda_h + \lambda \mathcal{I}_{\mathcal{H}})^{-1}} \cdot \\
&\quad\quad \mathcal{O}\left( H^2 \cdot G(n, 1 + 1/n) + \lambda \cdot R_r^2 + \zeta^2 \right)^{1/2},
\end{aligned}
$$

*where*

$$
\zeta = \mathcal{O}\left( d_{ef}^{samp} \sqrt{H \cdot \log\left( N(\mathcal{Q}, \|\cdot\|_\infty, 1/n)/\delta \right)} \cdot H e^H \right)
$$

*here $d_{ef}^{samp}$ is the sampling effective dimension.*

Here $\|\phi(z)\|_{(\Lambda_h + \lambda \mathcal{I}_{\mathcal{H}})^{-1}} = \langle \phi(z), (\Lambda_h + \lambda \mathcal{I}_{\mathcal{H}})^{-1} \phi(z) \rangle_{\mathcal{H}}$, where we define

$$
\Lambda_h = \sum_{i=1}^{n} \phi(z_h^i)\phi(z_h^i)^\top.
$$

With the guarantee of Theorem 5.3, Theorem 5.4 establishes the concrete suboptimality of DCPPO under various eigenvalue decay conditions.

**Theorem 5.4 (Suboptimality Gap for RKHS).** *Suppose that Assumption 5.1 holds. For $\mu$-polynomial decay, we further assume $\mu(1 - 2\tau) > 1$. For Algorithm 1 and 2, we set*

$$
\lambda = \begin{cases} C \cdot \mu \cdot \log(n/\delta) & \mu\text{-finite spectrum,} \\ C \cdot \log(n/\delta)^{1+1/\mu} & \mu\text{-exponential decay,} \\ C \cdot n^{\frac{2}{\mu(1-2\tau)-1}} \cdot \log(n/\delta) & \mu\text{-polynomial decay,} \end{cases}
$$

*and*

$$
\beta = \begin{cases} C'' \cdot H e^H |\mathcal{A}| d_{ef}^{samp} \cdot \sqrt{\mu \cdot \cdot \log(nH/\delta)} \\ C'' \cdot H e^H |\mathcal{A}| d_{ef}^{samp} \sqrt{(\log(nH/\delta))^{1+1/\mu}} \\ C'' \cdot H e^H |\mathcal{A}| d_{ef}^{samp} \cdot n^{\kappa^*} \cdot \sqrt{\log(nH/\delta)} \end{cases}
$$

*for $\mu$-finite spectrum, $\mu$-exponential decay and $\mu$-polynomial decay respectively. Here $C > 0$ is an absolute constant that does not depend on $n$ or $H$. Then with probability at least $1 - \delta$, it holds that*

$$
\begin{aligned}
&\text{SubOpt}(\{\tilde{\pi}_h\}_{h \in [H]}) \\
&\leq \begin{cases} C' \cdot \tilde{d} \cdot H e^H |\mathcal{A}| \sqrt{\mu \cdot \log(nR_r H/\delta)} \} \\ C' \cdot \tilde{d} \cdot H e^H |\mathcal{A}| \cdot \sqrt{(\log(nR_r H)/\delta)^{1+1/\mu}} \\ C' \cdot \tilde{d} \cdot H e^H |\mathcal{A}| \cdot (nR_r)^{\kappa^*} \cdot \sqrt{\log(nR_r H/\delta)}. \end{cases}
\end{aligned}
$$

*for $\mu$-finite spectrum, $\mu$-exponential decay and $\mu$-polynomial decay respectively. Here $C, C', C''$ are absolute constants irrelevant to $n$ and $H$ and*

$$
\tilde{d} = d_{ef}^{pop} \cdot d_{ef}^{samp}, \quad \kappa^* = \frac{d+1}{2(\mu+d)} + \frac{1}{\mu(1-2\tau)-1}.
$$

*Here $d_{ef}^{pop}$ is the population effective dimension, which measures the "coverage" of the human behavior $\pi_b$ for the optimal policy $\pi^*$.*

Parallel to the linear setting, Theorem 5.4 demonstrates the performance of our method in terms of effective dimension. If the behavior policy is close to the optimal policy and the RKHS satisfies Assumption 5.1, $d_{ef}^{pop} = \mathcal{O}(H^{3/2} n^{-1/2})$ and $d_{ef}^{samp}$ remains in constant level. In this case, suboptimality is of order $\mathcal{O}(n^{-1/2})$ for $\mu$-finite spectrum and $\mu$-exponential decay, while for $\mu$-polynomial decay we obtain a rate of $\mathcal{O}(n^{\kappa^*-1/2})$, which also matches the results in standard pessimistic planning under RKHS case (Jin et al., 2021), where the reward is observable.

## 6. Conclusion

In this paper, we have developed a provably efficient online algorithm, Dynamic-Choice-Pessimistic-Policy-Optimization (DCPPO) for RLHF under dynamic discrete choice model. By maximizing log-likelihood function of the Q-value function and minimizing mean squared Bellman error for the reward, our algorithm learns the unobservable reward, and the optimal policy following the principle of pessimism. We prove that our algorithm is efficient in sample complexity for linear model MDP and RKHS model class.

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
