# OpenReview forum: "Reinforcement learning with Human Feedback: Learning Dynamic Choices via Pessimism"
_ICML.cc/2023/Workshop/ILHF — ILHF Workshop ICML 2023_

### Official Review · Reviewer_GrMZ · 2023-06-16

**Rating:** 7
**Confidence:** 4

**Review:**

This paper studies Offline Reinforcement Learning with Human Feedback (RLHF) where we aim to learn the human’s underlying reward and the MDP’s optimal policy from a set of trajectories induced by human choices. It proposes a Dynamic-Choice-Pessimistic-Policy- Optimization (DCPPO) method and prove that the suboptimality of DCPPO almost matches the classical pessimistic offline RL algorithm in terms of suboptimality’s dependency on distribution shift and dimension. This is a good paper and has high quality. In the future, one can consider whether the variance structure can be recovered under the similar RLHF (for either linear or kernel regression).

---

### Official Review · Reviewer_hyeJ · 2023-06-16
**Well-written,**

**Rating:** 9
**Confidence:** 3

**Review:**

This is a well-written and easy to follow paper on RL from human feedback data. The authors provide theoretical guarantees for linear MDPs for an agent that first learns policies from given human-induced trajectories, then estimates the reward function that the policies optimize, and subsequently plans in a pessimistic way in the estimated environment. Overall I think the combination of the reward estimation process and pessimistic value iteration with the provided theoretical results is interesting and valuable and that the paper makes a good addition to the workshop.
A couple of things that were not very clear to me were: The difference between Dynamic Discrete Choice modelling and Max-Ent RL and what, if any, were the differences with the proposed reward estimation and IRL.

---

### Decision · Program_Chairs · 2023-06-20

Accept